# Responses in Nodulated Bean (*Phaseolus vulgaris* L.) Plants Grown at Elevated Atmospheric CO_2_

**DOI:** 10.3390/plants12091828

**Published:** 2023-04-29

**Authors:** Enrique Bellido, Purificación de la Haba, Eloísa Agüera

**Affiliations:** Department of Botany, Ecology and Plant Physiology, Faculty of Science, University of Córdoba, 14071 Córdoba, Spain; d92beage@uco.es (E.B.); bv1hahep@uco.es (P.d.l.H.)

**Keywords:** biofertilization, C:N ratio, carbon metabolites, N_2_ fixation, nitrogen metabolites

## Abstract

The increase in the concentration of CO_2_ in the atmosphere is currently causing metabolomic and physiological changes in living beings and especially in plants. Future climate change may affect crop productivity by limiting the uptake of soil resources such as nitrogen (N) and water. The contribution of legume–rhizobia symbioses to N_2_ fixation increases the available biological N reserve. Elevated CO_2_ (eCO_2_) has been shown to enhance the amount of fixed N_2_ primarily by increasing biomass. Greater leaf biomass under eCO_2_ levels increases N demand, which can stimulate and increase N_2_ fixation. For this reason, bean plants (*Phaseolus vulgaris* L.) were used in this work to investigate how, in a CO_2_-enriched atmosphere, inoculation with rhizobia (*Rhizobium leguminosarum*) affects different growth parameters and metabolites of carbon and nitrogen metabolism, as well as enzymatic activities of nitrogen metabolism and the oxidative state of the plant, with a view to future scenarios, where the concentration of CO_2_ in the atmosphere will increase. The results showed that bean symbiosis with *R. leguminosarum* improved N_2_ fixation, while also decreasing the plant’s oxidative stress, and provided the plant with a greater defense system against eCO_2_ conditions. In conclusion, the nodulation with rhizobia potentially replaced the chemical fertilization of bean plants (*P. vulgaris* L.), resulting in more environmentally friendly agricultural practices. However, further optimization of symbiotic activities is needed to improve the efficiency and to also develop strategies to improve the response of legume yields to eCO_2,_ particularly due to the climate change scenario in which there is predicted to be a large increase in the atmospheric CO_2_ concentration.

## 1. Introduction

The atmospheric CO_2_ concentration is the main driver for global climate change and has increased by 45% since the pre-industrial period, moving from 280 ppm to the current 406 ppm [1]. According to the RCP 8.5 scenario, the concentration of atmospheric CO_2_ could exceed 700 ppm by the end of this century [2]. The increase in CO_2_ encourages the photosynthesis of C3 crops and increases growth and yield through the ‘CO_2_ fertilization effect’ [3]. However, the initial stimulation of photosynthesis may decrease during long-term exposure to increased levels of CO_2_ concentration (eCO_2_), which is a phenomenon known as ‘photosynthetic acclimation’ [4]. Photosynthetic acclimation is related to the limited strength of sinks [5]. Sinks are organs/tissues that consume photosynthates, and their ability to consume and use photosynthates is called the sink strength [6]. High levels of CO_2_ concentration greatly increase the supply of carbohydrates, but if the sink strength is sufficient, the photosynthetic capacity may not be affected [7,8]. However, the decrease in the growth of the sink organ due to environmental factors may cause negative feedback for the photosynthetic capacity, resulting in photosynthetic acclimation for which the maximum rate of Rubisco carboxylation (*V_cmax_*) and the maximum rate of electron transport (*J*_max_) are deregulated by long-term exposure to eCO_2_ concentrations [3,9]. The continuous production or growth of organs, like the sustained growth of leaves, has been shown to minimize acclimation [10]. For legumes, not only shoots and roots, but also symbionts in nodules are important additional sinks [11]. The additional C sink from the nodules may help to avoid photosynthetic acclimation [12,13], which was shown in an experiment in which a non-nodulated soybean plants presented acclimation, however, a nodulated isogenic line did not acclimate [12,14] as a low level of eCO_2_ would increase sink strength and allow greater C export from source leaves [15]. In addition to the measured feedback from the source-sink, photosynthetic acclimation is also associated with a decrease in tissue N concentration, which is commonly observed in plants grown under eCO_2_ conditions [16]. It seems that the balance between the offer and demand of N during the growth and the balance between carbohydrate production and its use in the tissue and whole plant is crucial [3].

Beans and other legumes form symbiotic associations with bacteria that fix N_2_. For soybeans, it was found that symbiosis can consume from 4 to 11% of the fixed photosynthates, providing a large C sink that can be back-fed by stimulating additional photosynthetic fixation [17]. The efficiency of the biological fixation of N_2_ is modulated by several factors, with one of them being the physiological state of the host plant, which depends largely on environmental conditions [18]. The fixation of N_2_ is supplied by C, which is provided by the host plant and consequently, growth at eCO_2_ levels stimulates photosynthesis in C3 plants and can support enhanced N_2_ fixation in legumes [19]. It has been experimentally shown that the contribution of symbiotically fixed N_2_ to the total N of the plant increases eCO_2_ levels [20]. Some studies hypothesize that the productivity of legumes in response to eCO_2_ depends on the rhizobia strain used [21]. It has been shown that nodulation with strains that have a higher N_2_ fixation rate can avoid sink limitation in plants growing at eCO_2_ levels. Legumes that are N_2_ fixers may be able to decrease the C/N balance as they can additionally distribute fixed carbon to their symbionts in the nodules [15]. This not only leads to a larger carbohydrate sink but also stimulates N_2_ fixation that is synchronous with the carbohydrates offered [13,22].

At the global level, the biological fixation of N_2_ associated with annual grain legumes is approximately 21.5 million tons of N per year [23], which is a quarter of the annual input of nitrogen fertilizers used for cultivable land [24]. Therefore, the main objective of this work is to carry out a metabolomic and physiological study to determine whether bean (*P. vulgaris* L.) plants that have been inoculated with rhizobia (*R. Leguminosarum*) are able to increase the sink strength of the nodules at eCO_2_, thus allowing the legumes to maximize C and N_2_ fixation gains. In this way, we would be able to demonstrate how the presence of legumes in agricultural systems would decrease the need for chemical fertilizers and would provide economic and environmental benefits [25], particularly due to the climate change scenario in which there is predicted to be a large increase in atmospheric CO_2_ concentration.

## 2. Results

### 2.1. Growth Parameters

Non-nodulated (NN) and nodulated (N) plants grown under eCO_2_ conditions had higher biomass than those grown under aCO_2_ conditions. eCO_2_ and nodulation with rhizobia increased the leaf dry weight by approximately 30% (Table 1). In addition, it was also shown that the greatest increase in dry weight in relation to the area (specific leaf mass—SLM) was observed by the eCO_2_ and N plants (Table 1).

### 2.2. Leaf Sugar and C-N Content

In the bean plants (*P. vulgaris* L.), it was observed that when grown under eCO_2_ conditions, the glucose, fructose, sucrose, and maltose contents increased significantly as compared to plants grown under aCO_2_ conditions (*p* < 0.01) (Table 2). In addition, we can verify that when we inoculated with rhizobia, their carbon compound levels increased when grown under eCO_2_. Moreover, it can be seen that the glucose + fructose/sucrose ratio was higher in plants with eCO_2_ and N, which means that these plants contained a greater number of free monosaccharides than sugars in the form of sucrose (Table 2).

In bean leaves, we observed that the percentage of C in the leaf (leaf C%) did not vary between treatments; however, the percentage of N in the leaf (leaf N%) was lower in plants grown under eCO_2_ and NN conditions, although in these conditions, inoculation by rhizobia increased the percentage of N in the plant (Table 3). As a result, the C/N ratio was higher in plants grown under eCO_2_ and NN conditions, which showed that the inoculation with rhizobia (N) at eCO_2_ levels restored the low nitrogen levels caused by eCO_2_ (Table 3).

### 2.3. Leaf Amino Acid Contents

Free amino acid concentrations (glutamine, glutamate, asparagine, and aspartate) were measured (Table 4). eCO_2_ produced an accumulation of glutamine, glutamate, asparagine, and aspartate in NN and N plants compared to plants grown under aCO_2_ conditions (Table 4). An increase in the (Glu + Asp)/(Gln + Asn) ratio associated with nodulation in both aCO_2_ and eCO_2_ conditions was observed (Table 4), indicating that nodulated plants had a lower amide (Gln and Asn) content.

Table 5 shows the contents of the amino acids serine, proline, hydroxyproline, and GABA. It was shown that the levels of serine decreased at eCO_2_ levels (N and NN) with respect to aCO_2_ plants; however, it was also shown that nodulation caused a significant (*p* < 0.01) increase in serine levels in eCO_2_ plants with respect to eCO_2_ and NN plants. Moreover, it was found that proline, hydroxyproline, and GABA contents increased at eCO_2_ levels in N plants with respect to eCO_2_ and NN plants by 55.3%, 36.1%, and 71.4%, respectively. These results show that rhizobia inoculation increases the amino acids that can mitigate the stress caused by eCO_2_ in bean plants.

### 2.4. Enzyme Activities of Nitrogen Metabolism

The available evidence indicated that eCO_2_ concentrations alter the activity and expression of some enzymes that play a key role in nitrogen metabolism (nitrate reductase, NR, and glutamine synthetase, GS) in plants (Table 6). In fact, bean plants grown under eCO_2_ and NN conditions exhibited significantly (*p*< 0.01) lower NR and GS activities than those grown under aCO_2_ and NN conditions. However, it was observed that in nodulated (N) plants (aCO_2_ and eCO_2_), there was an increase in GS activity, with the increase being greater in eCO_2_ plants (Table 6).

### 2.5. Photosynthetic Pigment Contents

The plants grown under eCO_2_ conditions presented lower, statistically significant (*p* < 0.01) chlorophyll *a* and *b* and carotenoid contents than those grown under aCO_2_ conditions (Table 7). We observed that the total chlorophyll content decreased by 58.8% in NN plants and 43.1% in N plants grown under eCO_2_ conditions with respect to those grown at aCO_2_ levels (Table 7). In other words, the nodulation with rhizobia affected, to a lesser degree, the decrease in the photosynthetic pigment contents caused by eCO_2_. Similarly, the carotenoid content presented the same behavior, showing that the decrease in carotenoid content was greater in eCO_2_ NN (52.1%) than in eCO_2_ N (47.9%) plants, with respect to those grown in aCO_2_ conditions (Table 7).

### 2.6. Enzyme Activities of Antioxidant Systems and Hydrogen Peroxide Content

H_2_O_2_ content and antioxidant enzymes activities (catalase and ascorbate peroxidase, APX) in bean (*P. leguminosarum* L.) plant leaves were examined for both N and NN plants grown under eCO_2_ and aCO_2_ conditions. Table 8 shows that bean plants grown under eCO_2_ conditions had higher levels of H_2_O_2_ than those grown under aCO_2_ conditions (NN). However, when the plants were inoculated with *R. leguminosarum*, the hydrogen peroxide levels decreased by 25% in the aCO_2_ plants and by 60% in the eCO_2_ plants. We also noted an increase in catalase and APX activities in N plants with respect to NN plants, for both CO_2_ treatments (aCO_2_ and eCO_2_). From the results, we can say that nodulation with *R. leguminosarum* decreased the oxidative state of bean leaves.

## 3. Discussion

Our study aimed to examine the metabolic and physiological changes occurring in trifoliate bean leaves (*P. vulgaris* L.) when grown in an atmosphere with eCO_2_ and fixing N_2_. Elevated CO_2_ concentration is an important environmental factor affecting the rate of plant growth and development. Previous studies on sunflower plants showed that plants grown at eCO_2_ levels presented more pronounced growth than control plants [26,27,28]. Our study found that eCO_2_ increased the dry weight and leaf area of bean (*P. vulgaris* L.) leaves grown under nitrogen (NN) conditions (Table 1). We could observe that when plants grown at eCO_2_ were inoculated with *R. leguminosarum* ISP 14 (N), the dry weight increased with respect to NN plants by 30%. It can also be noted that there was a greater increase in dry weight with respect to leaf area (SLM) (Table 1). For soybean plants [29] grown at eCO_2_ levels, there was an increase in biomass when they were inoculated with *Brayrhizobium japonicum*; however, differences were observed depending on the strains used.

At eCO_2_ levels, there was an increase in monosaccharides (glucose and fructose) in bean plants (*P. vulgaris* L.) that was more significant when the plants were inoculated with *R. leguminosarum* (N) (Table 2). For some crops, sink limitation and photosynthesis downregulation are sometimes observed in plants grown at eCO_2_ levels as a consequence of sugar overaccumulation in leaves [9,30]. However, for legume crops that fix nitrogen, the symbionts can consume 4 to 11% of the carbohydrates fixed through photosynthesis and thereby increase the sink capacity of the plant, which can stimulate legume growth under eCO_2_ conditions and avoid C sink limitations [12]. It has been reported that an additional carbohydrate supply under eCO_2_ conditions leads to an increase in photosynthates as well as an increase in symbiotic N_2_ fixation with no additional N requirement [19,31]. For bean plants (*P. vulgaris* L.) grown with eCO_2_ and N, we could see that the glucose + fructose/sucrose ratio was significantly higher than in the rest of treatments, meaning that these plants had more free sugars in relation to sucrose (Table 2), which are available to the plants under these conditions.

Generally, plants growing at eCO_2_ levels are more nitrogen-limited than carbon-limited. It was shown that eCO_2_ modifies the N acquisition patterns of crops, for example, through limiting N uptake [32] or through the inhibition of NO_3_^−^ assimilation [33]. In addition, eCO_2_ reduces photorespiration, which limits energy transfer to NO_3_^−^ reduction and thereby NO_3_^−^ assimilation [34]. Leguminous plants obtain N through various pathways: (1) legumes uptake ammonia (NH_4_^+^) from the soil and incorporate it into organic compounds; (2) legumes uptake nitrate from the soil and reduce it to NH_4_^+^; and (3) legumes in symbiosis with N-fixing bacteria can obtain N from the atmosphere through N fixation and convert N_2_ into NH_4_^+^ [35]. Among these three pathways, N fixation is the costliest in terms of energy and resources. Acquiring N via the uptake of nitrate or ammonia from the soil requires fewer carbohydrates than acquiring N through symbiosis [36,37]. For sunflower plant leaves, it was observed that eCO_2_ increased the C/N ratio due to a lower N content in the plant [16]. Bellido et al. [28] showed that sunflower symbiosis with *Rhizophagus irregularis* improves the absorption of nitrogen, favoring the stability of the C:N ratio in plants when grown at eCO_2_ levels. In bean leaves, it was observed that the C:N ratio was higher in plants grown under eCO_2_ and NN conditions; however, nodulation with *R. leguminosarum* ISP 14 in combinated with eCO_2_ decreased the ratio significantly, showing that nodulation could reestablish the low nitrogen levels caused by eCO_2_ (Table 3). In *Pisum sativum* L. grown using a Free-Air CO_2_ Enrichment (FACE), it was found that under eCO_2_ conditions, there was a higher proportion of total N in the plant which resulted from N_2_ fixation and a small proportion of N taken up from the soil compared to plants grown under aCO_2_ conditions [38,39]. The lower availability of nitrogen in the leaves of NN bean (*P. vulgaris* L.) plants grown under eCO_2_ concentrations was related to significantly (*p* < 0.01) lower NR and GS activities than those grown under aCO_2_ (Table 6). Guo et al. [40] showed that eCO_2_ downregulated *NR* and *NT* but did not significantly affect the ammonia transporter *AMT*; thus, the decrease in N uptake from soil was mainly associated with the decrease in nitrate uptake rather than ammonia uptake. The decrease in nitrate uptake at eCO_2_ levels can be explained by a lower N availability in the soil and/or by lower nitrate assimilation [34,40]. When bean plants (*P. vulgaris.* L.) were inoculated with *R. Leguminosarum* there was a significant increase in GS activity in both aCO_2_ and eCO_2_ conditions, which was higher in plants grown at eCO_2_ levels. Nodulated plants takes up ammonia through the GS/GOGAT cycle, which reaches the leaves as ureides, thus increasing the nitrogenous compounds (glutamine, glutamate, asparagine, and aspartate) in the leaves of eCO_2_ and N plants compared to aCO_2_ and N plants as shown in Table 4. In alfalfa plants, it was observed that eCO_2_ increases nitrogen fixation, which coincides with a higher amount of amino acids and organic acids [41]. The Glu + Asp/Gln + Asn ratio showed that the ammonia content in the leaves in the form of amides was lower in N plants (Table 6), although this decrease was less marked in plants grown at eCO_2_ levels. Soba et al. [29] noted that under eCO_2_ conditions in the nodules, there is a more efficient use of C and N and this happens through carboxylation of PEP to produce aspartate, rather than through a decarboxylation of PEP to produce dicarboxylic acids from the Krebs cycle. As such, the fixation of CO_2_ in the nodules may represent a C and aspartate accumulation mechanism that can be used to export N or produce ureides to be exported to the leaves. Parvin et al. [42] showed that in *Lens culinaris* L. inoculated with *R. leguminosarum*, eCO_2_ led to a more accelerated leaf N mobilization.

Bean *(P. vulgaris* L.) plants were grown under eCO_2_ conditions and the content of photosynthetic pigments (chlorophyll *a* and *b* and carotenoids) decreased compared to plants grown under aCO_2_ conditions (Table 7). Similar results were observed for sunflower plant leaves growing under the same conditions [27,28]. However, the significant increase in the chlorophyll and carotenoid levels that were found in nodulated plants grown under eCO_2_ conditions may be related to the increased N content in these plants (Table 3). Sanz-Sáez et al. [43] did not observe an increase in leaf N and chlorophyll contents, both of which are an indicator of nitrogen fixation under field conditions, due to the low effectiveness of the inoculation. Carotenoids act as light-harvesting pigments and play a major role in protecting chlorophyll and membranes from destruction by quenching triplet chlorophyll and removing oxygen from the excited chlorophyll–oxygen complex [44]. Therefore, the reduction in carotenoids in eCO_2_ and NN plants and the increase in levels when plants are inoculated with *R. leguminosarum* (eCO_2_ and N) may have major consequences in terms of chlorophyll behavior.

In plants grown under eCO_2_ conditions, oxidative stress was increased, favoring the production of ROS, which can be seen through the high level of hydrogen peroxide contained in the leaves compared to plants grown under aCO_2_ conditions [27,28]. In our study, we observed an increase in hydrogen peroxide levels in leaves when bean plants were grown at eCO_2_ compared to those grown at aCO_2_ levels (Table 8). However, when the plants were inoculated with rhizobia (*R. leguminosarum*), there was an increase in the levels of catalase and APX and a significant decrease in hydrogen peroxide content in both eCO_2_ and aCO_2_ conditions (Table 8). The decrease in catalase and APX activities in eCO_2_ plants may be related to the observed reduction in the N level (Table 3). ROS can be dangerous for biological processes and structures and can result in the oxidation of DNA, amino acids, and proteins, as well as lipid peroxidation. In order to prevent the detrimental effects of ROS, plants have developed a robust antioxidant defense system that decreases the damage from free radicals [45]. Surprisingly, oxidative stress and the antioxidant system may be changed in a future climate [13,27] and increased levels of antioxidant molecules and enzymes are associated with tolerance to stress [45]. N deprivation leads to senescence in sunflower plant leaves [16,27] and has also been shown in soybean to increase phytol, free fatty acids, and other compound levels related to chlorophyll and membrane degradation that may be caused by oxidative stress due to the low nitrogen levels [29]. Photorespiration maintains redox homeostasis within plant cells because it can dissipate many potentially dangerous compounds [46]. For bean plants grown at eCO_2_ levels, serine levels decreased and the effect was more marked in NN plants (Table 5). It has been found that when plants are grown under eCO_2_ conditions, photorespiration decreases [46]. Moreover, we can also confirm that the amino acids proline, hydroxyproline, and GABA (Table 5) increased under eCO_2_ conditions in both NN and N plants. These amino acids are related to stress tolerance [47], and therefore we can confirm that bean plants grown under eCO_2_ conditions and nodulated with *R. leguminosarum* show greater tolerance to the stress caused by an atmosphere that has a high CO_2_ content since it provides the plant with a greater defense system in the face of eCO_2_ conditions [29,31,42]. Lopez et al. [48] found N_2_-fixing bean plants were more protected than those fertilized with nitrate when grown under drought conditions as increased levels of ABA, proline, and amino acids were observed.

## 4. Materials and Methods

### 4.1. Plant Material, Growth Conditions, and Experimental Design

This work examined modifications in the development and metabolism of the common bean (*Phaseolus vulgaris* L. cv. Great Northern) grown under enriched CO_2_ conditions (eCO_2_) and inoculated with *Rhizobium leguminosarum* ISP14 (courtesy of Dr. Dulcenombre Rodriguez C.I.F.A., Sevilla, Spain) (N), compared to the control bean grown under eCO_2_ conditions with no inoculation (NN). We also used control plants grown under ambient CO_2_ conditions (aCO_2_) with N and NN. The seeds (*P. vulgaris* L. cv. Great Northern), provided by Professor A. De Ron (CSIS Experimental Mission; Santiago de Compostela, Spain), were surface sterilized in 1% (*v*/*v*) hypochlorite solution for 15 min. They were subsequently placed in Petri dishes (120 mm in diameter) and, after three days, four seedlings were sown in plastic trays (16 cm in diameter, 18 cm in height) containing a 2:1 (*v*/*v*) mixture of perlite and vermiculite. The seedlings were inoculated with a fresh suspension of *R. leguminosarum* ISP14 that was incubated at a low temperature (28 °C) for less than 30 h. The seeds were germinated and the plants were grown in controlled environment cabinets (Sanyo Gallenkam Fitotron, Leicester, UK) fitted with an ADC 2000 CO_2_ gas monitor with a 16 h photoperiod (300 mol/m^2^/s) of photosynthetically active radiation supplied by “cool white” fluorescent lamps, supplemented by incandescent bulbs, and a day/night regime of 23/19 °C and 70/80% relative humidity. The NN plants were given a nutrient solution [49] three times a week and the N plants were given the same solution, but free of nitrogen. Samples of leaves (aged 28 days; first trifoliate leaf) were collected 2 h after the onset of the photoperiod. Whole leaves were excised and pooled into two groups. One group was used to take growth parameters. The other group was immediately frozen in liquid nitrogen and stored at –80 °C. The frozen plant material was ground in a mortar that was pre-cooled with liquid N_2_, and the resulting powder was distributed into small vials that were stored at –80 ºC until they were used for assays of enzyme activity and metabolite quantification.

### 4.2. Growth Parameters

Leaf dry weight was determined after drying the plant material in an oven at 80 °C until the weight was constant. Leaf area (image analysis software, Image-Pro Plus) measurements were taken to calculate specific leaf mass (SLM) in mg dry weight/cm^2^.

### 4.3. Carbohydrate and Amino Acid Determination in Leaves

Sugars and amino acids were analyzed following the procedure based on the Fiehn method [50] using gas chromatography-mass spectrometry and combined targeted and untargeted profiling (Agilent 7890B GC System combined with a LECO Pegasus HT High Throughput TOF-MS detector, Santa Clara, CA 95051 US).

### 4.4. Carbon and Nitrogen Determination in Leaves

This simultaneous C and N analysis required high-temperature combustion in an oxygen-rich environment and was based on the classical Dumas method. The combustion products are swept out of the combustion chamber by an inert carrier gas (helium) and passed over heated high-purity copper, to remove any oxygen not consumed in the initial combustion and to convert any oxides of nitrogen gas. The gases can be detected by gas chromatography (GC) separation followed by quantification using thermal conductivity detection (TCD) of individual compounds. The quantification of the elements requires calibration for each element using high-purity analytical standard compounds [51].

### 4.5. Determination of Pigments and H_2_O_2_ in Leaves

The pigments were measured in leaves extracts using HPLC, according to the protocol of Cabello et al. [52]. For H_2_O_2_ determination, 1 g of leaf material was ground with 10 mL cold acetone in a cold room and was passed through a Whatman filter paper. H_2_O_2_ was determined by the formation of a titanium–hydroperoxide complex in accordance with the method of Mukherjee and Choudhuri [53].

### 4.6. Extraction and Activity of Enzymes Involved in Nitrogen Metabolism in Leaves

Frozen material was homogenized in a chilled extraction medium (4 mL/g) consisting of 100 mM Hepes-KOH (pH 7.5), 10% (*v*/*v*) glycerol, 1% (*w*/*v*) polyvinylpolypyrrolidone (PVPP), 0.1% (*v*/*v*) Triton X-100, 6 mM dithiothreitol (DTT), 1 mM EDTA, 0.5 mM phenylmethylsulfonyl fluoride (PMSF), 25 μM leupeptin, 20 μM flavin adenine dinucleotide (FAD), and 5 μM Na_2_MoO_4_. The homogenate was centrifuged at 8000× *g* at 4 °C for 2 min, and enzyme activities (NR and GS) were measured immediately using the cleared extract. NR (E.C. 1.6.6.1) activity was determined in the absence of Mg^2+^ to measure the total NADH-NR activity. The nitrite formed was determined spectrophotometrically at 540 nm, following the method of Agüera et al. [54]. GS (E.C. 6.3.1.2) activity was measured using the transferase assay in a reaction mixture containing, in a final volume of 1 mL, 50 mM Hepes–KOH (pH 7.5), 30 mM L-glutamine, 60 mM NH_2_OH, 0.4 mM ADP, 3 mM MnCl_2_, 20 mM Na_2_HASO_4,_ and an adequate amount of enzyme preparation. The mixtures were incubated at 30 °C and the reactions were terminated by the addition of 2 mL of cold ferric chloride reagent (120 mM FeCl_3_, 78 mM HCl, and 73 mM trichloroacetic acid). The γ-glutamyl hydroxamate formed was determined spectrophotometrically at 500 nm, following the protocol of De la Haba et al. [55].

### 4.7. Extraction and Assay of Antioxidant Enzymes in Leaves

Enzyme extracts were prepared by freezing a weighed amount of leaf samples in liquid nitrogen to prevent proteolytic activity, followed by grinding in a 0.1 M phosphate buffer at pH 7.5 containing 0.5 mM EDTA and 1 mM ascorbic acid at a 1:10 (*w*/*v*) ratio. The homogenate was passed through four layers of gauze, and the filtrate was centrifuged at 15,000× *g* for 20 min. The resulting supernatant was used as an enzyme source. Catalase activity (CAT, E.C.1.11.1.6) was estimated using the method of Aebi [56]. The reaction mixture contained 50 mM potassium phosphate (pH 7) and 10 mM H_2_O_2_. After the enzyme was added, H_2_O_2_ decomposition was monitored via absorbance at 240 nm (ε = 43.6/(mM/cm)). Ascorbate peroxidase activity (APX, E.C.1.11.1.11) was measured using Nakano and Asada’s method [57]. The reaction mixture contained 50 mM phosphate buffer (pH 7), 1 mM sodium ascorbate, and 25 mM H_2_O_2_. After the addition of the enzymatic extract to the mixture, the reaction was monitored via absorbance at 290 nm (ε = 2.8/(mM/cm)).

### 4.8. Statistical Analysis

The data were subjected to a two-way ANOVA (inoculation with *R. leguminosarum* and CO_2_ level). Pairwise comparisons of means were performed using Turkey’s test, and statistically significant differences were obtained at *p* < 0.05. The results are presented as the mean ± SE of three independent experiments, performed sequentially, using duplicate determinations in each experiment.

## 5. Conclusions

Bean plants (*P. vulgaris* L.) in a CO_2_-enriched atmosphere (eCO_2_) were used to examine the effects of nodulation with *R. leguminosarum* (N) on a physiological and metabolic level. This symbiosis was found to promote plant growth and favor a greater synthesis of photosynthetic pigments. Nodulated plants (N) under eCO_2_ conditions had a higher concentration of carbon compounds in their leaves, compared to non-nodulated (NN) and eCO_2_ plants. For eCO_2,_ the nodulation (N) of beans with *R. leguminosarum* decreased the C:N ratio compared to the NN plants, and also decreased the hydrogen peroxide content and increased the antioxidant enzyme activity (catalase and APX). These results suggest that bean symbiosis with *R. leguminosarum* improves the absorption of N while also decreasing the plant’s oxidative stress, and gives the plant a better defense system against eCO_2_ conditions. Our results confirmed that the ability to increase the sink strength of nodules is an important mechanism that allows legumes to maximize C and N_2_ fixation gains in a future high-CO_2_ atmosphere. In conclusion, the nodulation (biofertilization) with rhizobia (*R. leguminosarum*) may potentially replace the chemical fertilization of bean plants (*P. vulgaris* L.), resulting in more environmentally friendly agricultural practices. Therefore, the presence of legumes in agricultural systems decreases the need for chemical fertilizers, providing economic and environmental benefits [25]. However, further optimization of symbiotic activities is needed to improve the efficiency and yield of crop resource use [58], and to also develop strategies to improve the response of legume yields to eCO_2_, particularly due to the climate change scenario in which there is predicted to be a large increase in atmospheric CO_2_ concentration.

## Figures and Tables

**Table 1 plants-12-01828-t001:** Growth parameters of bean (*P. vulgaris* L.) plants, nodulated (N) or non-nodulated (NN) with *R. leguminosarum*, grown in ambient (aCO_2_) and elevated (eCO_2_) conditions. Data are shown as means ± SE. Different letters show significant differences among the treatments according to Tukey’s test (*p* < 0.05). ** *p* < 0.01, ** p* < 0.05.

		Leaf Dry Weight	Leaf Area	SLM
		(mg)	(cm^2^)	(mg DW/cm^2^)
aCO_2_				
	NN	173.3 ± 20	231.45 ± 36.26	0.77 ± 0.20
	N	155.5 ± 10	227.67 ± 30.74	0.70 ± 0.06
eCO_2_				
	NN	217.3 ± 20	341.53 ± 3.89	0.64 ± 0.04
	N	282.22 ± 10	260.06 ± 0.70	1.09 ± 0.03
Source of variation				
N		**	*	*
CO_2_		**	**	**
N × CO_2_		****	***	****

**Table 2 plants-12-01828-t002:** Glucose, fructose, sucrose, and maltose contents in the leaves of bean (*P. vulgaris* L.) plants, nodulated (N) or non-nodulated (NN) with *R. leguminosarum*, grown in ambient (aCO_2_) and elevated (eCO_2_) conditions. Data are shown as means ± SE. Different letters show significant differences among the treatments according to Tukey’s test (*p* < 0.05). ** *p* < 0.01.

		Glucose	Fructose	Sucrose	Maltose	(Glucose + Fructose)/Sucrose
		(mg g^−1^ DW)	(mg g^−1^ DW)	(mg g^−1^ DW)	(mg g^−1^ DW)	Ratio
aCO_2_						
	NN	116.99 ± 0.23	52.35 ± 0.14	282.67 ± 0.10	4.75 ± 0.07	0.60
	N	69.54 ± 0.24	125.97 ± 0.16	167.49 ± 0.28	2.39 ± 0.02	1.17
eCO_2_						
	NN	334.08 ± 0.61	84.58 ± 0.87	378.42 ± 0.17	4.97 ± 0.02	1.11
	N	570.10 ± 0.71	212.82 ± 0.50	272.49 ± 0.54	8.52 ± 0.03	2.69
Source of variation						
N		**	****	**	****	
CO_2_		**	****	**	****	
N × CO_2_		****	****	****	****	

**Table 3 plants-12-01828-t003:** Contents of carbon and nitrogen and the C:N ratio in the leaves of bean (*P. vulgaris* L.) plants, nodulated (N) or non-nodulated (NN) with *R. leguminosarum*, grown in ambient (aCO_2_) and elevated (eCO_2_) conditions. Data are shown as means ± SE. Different letters show significant differences among the treatments according to Tukey’s test (*p* < 0.05). ** *p* < 0.01, ** p* < 0.05, *NS* = not significant.

		Leaf C	Leaf N	C:N
		(%)	(%)	Ratio
aCO_2_				
	NN	41.29 ± 1.49	2.12 ± 0.43	19.61
	N	42.29 ± 0.46	2.84 ± 0.19	14.89
eCO_2_				
	NN	41.59 ± 1.97	1.67 ± 0.18	24.90
	N	43.15 ± 0.20	3.06 ± 0.72	14.10
Source of variation				
N		*NS*	*	**
CO_2_		*NS*	*	*NS*
N × CO_2_		*NS*	*	*NS*

**Table 4 plants-12-01828-t004:** Glutamine, glutamate, asparagine, and aspartate contents in the leaves of bean plants, nodulated (N) or non-nodulated (NN) with *R. leguminosarum*, grown in ambient (aCO_2_) and elevated (eCO_2_) conditions. Data are shown as means ± SE. Different letters show significant differences among the treatments according to Tukey’s test (*p* < 0.05). *** p* < 0.01.

		Glutamine	Glutamate	Asparagine	Aspartate	(Glu + Asp/Gln + Asn)
		(mg g^−1^ DW)	(mg g^−1^ DW)	(mg g^−1^ DW)	(mg g^−1^ DW)	Ratio
aCO_2_						
	NN	3.22 ± 0.21	1.43 ± 0.03	1.66 ± 0.02	10.89 ± 0.29	2.52
	N	0.88 ± 0.01	4.97 ± 0.19	0.86 ± 0.03	11.06 ± 0.38	9.21
eCO_2_						
	NN	3.33 ± 0.19	21.64 ± 0.26	17.23 ± 0.01	15.32 ± 0.50	1.69
	N	1.97 ± 0.15	17.8 0 ± 0.46	3.73 ± 0.15	13.19 ± 0.16	5.55
Source of variation						
N		**	****	**	****	
CO_2_		**	****	**	****	
N × CO_2_		****	****	****	****	

**Table 5 plants-12-01828-t005:** Serine, proline, hydroxyproline, and GABA contents in the leaves of bean (*P. vulgaris* L.) plants, nodulated (N) or non-nodulated (NN) with *R. leguminosarum*, grown in ambient (aCO_2_) and elevated (eCO_2_) conditions. Data are shown as means ± SE. Different letters show significant differences among the treatments according to Tukey’s test (*p* < 0.05). ** *p* < 0.01, *NS* = not significant.

		Serine	Proline	Hydroxyproline	GABA
		(mg g^−1^ DW)	(mg g^−1^ DW)	(mg g^−1^ DW)	(mg g^−1^ DW)
aCO_2_					
	NN	2.88 ± 0.12	1.15 ± 0.04	0.58 ± 0.03	2.34 ± 0.09
	N	2.56 ± 0.05	1.14 ± 0.04	0.57 ± 0.01	2.71 ± 0.08
eCO_2_					
	NN	1.96 ± 0.02	1.21 ± 0.01	0.68 ± 0.01	2.59 ± 0.05
	N	2.23 ± 0.17	1.88 ± 0.12	0.93 ± 0.02	4.44 ± 0.34
Source of variation					
N		*NS*	****	**	****
CO_2_		**	****	**	****
N × CO_2_		****	****	****	****

**Table 6 plants-12-01828-t006:** Nitrate reductase and glutamine synthetase activities in the leaves of bean (*P. vulgaris* L.) plants, nodulated (N) or non-nodulated (NN) with *R. leguminosarum*, grown in ambient (aCO_2_) and elevated (eCO_2_) conditions. Data are shown as means ± SE. Different letters show significant differences among the treatments according to Tukey’s test (*p* < 0.05). ** *p* < 0.01.

		NR	GS
		(µmol NO^−2^ h^−1^ g^−1^ DW)	(mmol γ-gh h^−1^ g^−1^ DW)
aCO_2_			
	NN	63.7 ± 1.80	114.72 ± 1.40
	N	49.07 ± 2.01	120.78 ± 2.05
eCO_2_			
	NN	29.67 ± 1.20	50.01 ± 1.97
	N	26.29 ± 1.81	200.13 ± 3.98
Source of variation			
N		**	****
CO_2_		**	****
N × CO_2_		****	****

**Table 7 plants-12-01828-t007:** Chlorophyll a, chlorophyll b, and carotenoid content in the leaves of bean (*P. vulgaris* L.), plants nodulated (N) or non-nodulated (NN) with *R. leguminosarum*, grown in ambient (aCO_2_) and elevated (eCO_2_) conditions. Data are shown as means ± SE. Different letters show significant differences among the treatments according to Tukey’s test (*p* < 0.05). ** *p* < 0.01, ** p* < 0.05, *NS* = not significant.

		Chlorophyll *a*	Chlorophyll *b*	Total Chlorophyll Content	Chlorophyll *a/b*	Carotenoids
		(mg g^−1^ DW)	(mg g^−1^ DW)	(mg g^−1^ DW)	Ratio	(mg g^−1^ DW)
aCO_2_						
	NN	37.29 ± 3.43	15.88 ± 2.28	53.18 ± 5.68	4.31 ± 0.40	7.36 ± 0.56
	N	37.99 ± 12.90	16.88 ± 6.67	54.87 ± 19.34	4.19 ± 1.95	7.46 ± 2.14
eCO_2_						
	NN	15.64 ± 2.63	6.35 ± 0.73	21.99 ± 3.33	3.32 ± 0.31	3.31 ± 0.9
	N	21.56 ± 3.20	9.64 ± 0.68	31.20 ± 3.87	2.340 ± 0.31	3.88 ± 1.01
Source of variation						
N		*NS*	*NS*	*NS*	*NS*	*NS*
CO_2_		**	**	**	*	**
N × CO_2_		*NS*	*NS*	*NS*	*NS*	*NS*

**Table 8 plants-12-01828-t008:** H_2_O_2_ content, catalase, and APX activities in the leaves of bean (*P. vulgaris* L.) plants, nodulated (N) or non-nodulated (NN) with R. *leguminosarum*, grown in ambient (aCO_2_) and elevated (eCO_2_) conditions. Data are shown as means ± SE. Different letters show significant differences among the treatments according to Tukey’s test (*p* < 0.05). ** *p* < 0.01, *NS* = not significant.

		H_2_O_2_	Catalase	APX
		(mg g^−1^ DW)	(U g^−1^ DW)	(Ug^−1^ DW)
aCO_2_				
	NN	21.51 ± 2.20	9.30 ± 0.15	122.99 ± 17.15
	N	15.14 ± 1.46	10.67 ± 1.55	129.4 ± 14.85
eCO_2_				
	NN	58.10 ± 3.55	7.10 ± 0.80	68.46 ± 4.38
	N	26.33 ± 3.04	8.85 ± 0.39	90.9 ± 15.50
Source of variation				
N		**	**	*NS*
CO_2_		**	**	****
N × CO_2_		**	*NS*	*NS*

## Data Availability

This research was funded by the University of Córdoba Programa Propio (XXPP.MO 4.1) and Junta de Andalucía PAI Group BIO-159.

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
