# Peer review of "Responses in Nodulated Bean (Phaseolus vulgaris L.) Plants Grown at Elevated Atmospheric CO2"

_plants, 2023, doi:10.3390/plants12091828_

Round 1
Reviewer 1 Report
The manuscript entitled “Metabolomic and Physiological Responses in Nodulated Bean (Phaseolus vulgaris L.) Plants Grown at Elevated Atmospheric CO2” described the effects of inoculation with rhizobia on bean performances under elevated CO2 condition. The work is interesting and also has a certain meaning.
Line 14: biomass of what?
Line 19: These results? The authors only described the purpose of the study before. The authors can revise it as “The results showed that”
Line 22: There are two “may” in one sentence.
The authors emphasize “Nodulated” in the title, but this word never appeared in the manuscript. Is there any difference between nodulated bean and normal beans? If yes, please describe it in the Introduction, If no, it is suggested to delete this word in the title.
Authors should be careful of the tense of the paper, when should you use present indefinite tense and when should you past tense?
Should the table notes be placed behind the table?
Tables: The decimal points should be uniform. The zero cannot be omitted.
The writing of the paper can be improved.
Author Response
If there are differences between nodulated and non-nodulated plants.
-The term has been introduced in the legend of the tables (Table 1, 2, 3, 4.5, 6, 7, and 8,) and the text and in various places for clarity:
Lines 66,67: “It has been shown that nodulation”.
Line 85: “Plants grown under eCO2 conditions and not nodulated (NN) and nodulated (N)”.
Line 260: “however, nodulation with R. leguminosarum ISP 14”.
Line 326,327: “eCO2 conditions and nodulated with R. leguminosarum”
Authors should be careful of the tense of the paper, when should you use present indefinite tense and when should you past tense?
It has been taken into account and has been modified from different places putting everything in Past Tense
Should the table notes be placed behind the table?
According to the format rules of Plants magazine, they indicate that they must be placed before.
Tables: The decimal points should be uniform. The zero cannot be omitted.
All missing zeros and periods have been placed as indicated: Table 1, 2, 6, 7 and 8
The writing of the paper can be improved.
The paper has been revised
An English-speaking person with scientific experience proofreads the manuscript to improve its readability. She has corrected the grammar, spelling and punctuation and has unified the entire text into British English and removed any American English words, using her broad knowledge of English language and usage to point out passages that were confusing or vague. Every effort has been made to ensure that neither the research content nor the authors' intentions were altered in any way during the proofreading process.

Reviewer 2 Report
Article ‘Metabolomic and …… CO2’ explains metabolomic and physiological responses of bean under high CO2 stress conditions. My concerns are:–
1. Choice of plant: Phaseolus vulgaris is well known for N2-fixation. Why this plant was selected for elevated CO2 study? For such type of studies, best plants are C-4 plants and may compared with C-3 plants.
2. Rhizobia are commonly involved in N-fixation, while in this study, it supports CO2 stress tolerance and increase photosynthesis. Any mechanistic literature or reports are there to support this concept?
3. Title says Metabolomics, therefore I expect a metabolomic study (GC-MS/LC-MS or NMR based), but they only study sugars and amino-acids. Why not non-targeted metabolomics was performed as mentioned in the title?
4. There is no physiological response study such as EL, MSI, different Plant growth parameters and also stress response indicators?
Author Response
Response to Reviewer 2 Comments
The authors would like to thank the Reviewer for his useful comments and suggestions. An itemized reply to his comments and queries follows.
- Choice of plant: r is well known for N2-fixation. Why this plant was selected for elevated CO2 study? For such type of studies, best plants are C-4 plants and may compared with C-3 plants.
We have chosen Phaseolus vulgaris precisely because it is a C3 plant since C4 plants lack photorespiration and the effect of CO2 is less than in C3. Our team has as a general research topic, C3 plants with different types of symbiosis seeing the effect of climate change. However, it would be interesting, as he indicates, later to compare with C4 plants.
- Rhizobia are commonly involved in N-fixation, while in this study, it supports CO2 stress tolerance and increase photosynthesis. Any mechanistic literature or reports are there to support this concept?
We have introduced in line 329 these three bibliographical references [30, 32, 43], to support this concept.
- Title says Metabolomics, therefore I expect a metabolomic study (GC-MS/LC-MS or NMR based), but they only study sugars and amino-acids. Why not non-targeted metabolomics was performed as mentioned in the title?
We have made a metabolomic study of the metabolites that are most affected by high CO2 and we have used other necessary techniques to analyze what was intended. Perhaps in future studies it will be done, in any case we will remove it from the title and it will be changed as “Responses in Nodulated Bean (Phaseolus vulgaris L.) Plants Grown at Elevated Atmospheric CO2”
- There is no physiological response study such as EL, MSI, different Plant growth parameters and also stress response indicators?
We have presented SLM growth parameters as well as stress indicators including proline and GABA as well as antioxidant enzymes (APX and Catalases) and reactive oxygen species (H2O2). Thank you for the indication to do EL and MSI will be done in later studies.”
An English-speaking person with scientific experience proofreads the manuscript to improve its readability. She has corrected the grammar, spelling and punctuation and has unified the entire text into British English and removed any American English words, using her broad knowledge of English language and usage to point out passages that were confusing or vague. Every effort has been made to ensure that neither the research content nor the authors' intentions were altered in any way during the proofreading process.
Response to Reviewer 2 Comments
The authors would like to thank the Reviewer for his useful comments and suggestions. An itemized reply to his comments and queries follows.
- Choice of plant: r is well known for N2-fixation. Why this plant was selected for elevated CO2 study? For such type of studies, best plants are C-4 plants and may compared with C-3 plants.
We have chosen Phaseolus vulgaris precisely because it is a C3 plant since C4 plants lack photorespiration and the effect of CO2 is less than in C3. Our team has as a general research topic, C3 plants with different types of symbiosis seeing the effect of climate change. However, it would be interesting, as he indicates, later to compare with C4 plants.
- Rhizobia are commonly involved in N-fixation, while in this study, it supports CO2 stress tolerance and increase photosynthesis. Any mechanistic literature or reports are there to support this concept?
We have introduced in line 329 these three bibliographical references [30, 32, 43], to support this concept.
- Title says Metabolomics, therefore I expect a metabolomic study (GC-MS/LC-MS or NMR based), but they only study sugars and amino-acids. Why not non-targeted metabolomics was performed as mentioned in the title?
We have made a metabolomic study of the metabolites that are most affected by high CO2 and we have used other necessary techniques to analyze what was intended. Perhaps in future studies it will be done, in any case we will remove it from the title and it will be changed as “Responses in Nodulated Bean (Phaseolus vulgaris L.) Plants Grown at Elevated Atmospheric CO2”
- There is no physiological response study such as EL, MSI, different Plant growth parameters and also stress response indicators?
We have presented SLM growth parameters as well as stress indicators including proline and GABA as well as antioxidant enzymes (APX and Catalases) and reactive oxygen species (H2O2). Thank you for the indication to do EL and MSI will be done in later studies.”
An English-speaking person with scientific experience proofreads the manuscript to improve its readability. She has corrected the grammar, spelling and punctuation and has unified the entire text into British English and removed any American English words, using her broad knowledge of English language and usage to point out passages that were confusing or vague. Every effort has been made to ensure that neither the research content nor the authors' intentions were altered in any way during the proofreading process.
If there are differences between nodulated and non-nodulated plants.
-The term has been introduced in the legend of the tables (Table 1, 2, 3, 4.5, 6, 7, and 8,) and the text and in various places for clarity:
Lines 66,67: “It has been shown that nodulation”.
Line 85: “Plants grown under eCO2 conditions and not nodulated (NN) and nodulated (N)”.
Line 260: “however, nodulation with R. leguminosarum ISP 14”.
Line 326,327: “eCO2 conditions and nodulated with R. leguminosarum”
Authors should be careful of the tense of the paper, when should you use present indefinite tense and when should you past tense?
It has been taken into account and has been modified from different places putting everything in Past Tense
Should the table notes be placed behind the table?
According to the format rules of Plants magazine, they indicate that they must be placed before.
Tables: The decimal points should be uniform. The zero cannot be omitted.
All missing zeros and periods have been placed as indicated: Table 1, 2, 6, 7 and 8
The writing of the paper can be improved.
The paper has been revised
An English-speaking person with scientific experience proofreads the manuscript to improve its readability. She has corrected the grammar, spelling and punctuation and has unified the entire text into British English and removed any American English words, using her broad knowledge of English language and usage to point out passages that were confusing or vague. Every effort has been made to ensure that neither the research content nor the authors' intentions were altered in any way during the proofreading process.

Round 2
Reviewer 1 Report
It can be seen that authors answered my comments and revised the manuscript.
Author Response
The authors would like to thank the Reviewer for his useful comments and suggestions
Reviewer 2 Report
Revision is satisfactory
Author Response

(The authors gave the same response as above.)
